# Development of a Best Practice Guidance on Online Peer Support for People with Young-Onset Dementia

**DOI:** 10.3390/bs14090746

**Published:** 2024-08-26

**Authors:** Esther Vera Loseto-Gerritzen, Orii McDermott, Martin Orrell

**Affiliations:** Institute of Mental Health, Mental Health and Clinical Neuroscience, School of Medicine, University of Nottingham, Nottingham NG7 2TU, UK; orii.mcdermott@nottingham.ac.uk (O.M.); martin.orrell@nottingham.ac.uk (M.O.)

**Keywords:** young-onset dementia, peer support, social support, online support, social health

## Abstract

This work aimed to develop a Best Practice Guidance on online peer support for people with young-onset dementia (YOD). The Best Practice Guidance was developed through a systematic literature review, focus groups, an online survey, and interviews and consultations with people with YOD and professionals. The Best Practice Guidance consists of two parts. Part 1 contains information for people with YOD about what online peer support entails, what to expect from it, and how to get involved. Part 2 is aimed at those who facilitate or moderate online peer support (professionals or people with lived experience) and includes guidelines on how to optimize online peer support for people with YOD. The Best Practice Guidance on online peer support provides (1) people with YOD with evidence-based, relevant, and accessible information about what online peer support entails and how it could help them, (2) providers and facilitators with guidelines on how to optimize online peer support for people with YOD, and (3) healthcare professionals with a concise and accessible tool for signposting. Future research is needed to implement and disseminate the Best Practice Guidance among dementia organizations and healthcare practices and should include rigorous studies on the implementation and sustainability of online peer support for people with YOD.

## 1. Introduction

Approximately 3.9 million people are living with young-onset dementia (onset before the age of 65 [1]) globally [2], of whom an estimated 70,800 are living in the United Kingdom (UK) [3]. As dementia is commonly associated with older age, a diagnosis in mid-life tends to be unexpected and can significantly disrupt someone’s life [4,5,6]. People with YOD and their families face unique challenges compared to older adults with dementia, due to the impact the diagnosis can have on employment [7] and roles and responsibilities within their families [8]. This is often accompanied by stigma and a lack of understanding from people’s social networks, healthcare professionals, and the wider society. These experiences can have a significant impact on a person’s sense of self and their identity [9,10]. 

Peer support can help with the many challenges that come with a YOD diagnosis. The importance and benefits of peer support for people with YOD have been widely researched [4,5,11], and people with YOD can see peer support as a positive post-diagnostic support service [12]. Peer support can provide a non-judgmental environment in which people can exchange support with those who are in a comparable situation. Peer support can help people feel less alone in their experiences and provide opportunities for social engagement, exchanging experiences and information, and new activities [6,13]. However, research shows that the availability of specialized YOD services varies widely across the UK [14]. As a result, many people with YOD experience difficulties in accessing local, age-appropriate support services, including peer support groups, and are left without much-needed post-diagnostic support. Online peer support could be a solution, as it allows people to take part independently of where they are, from the comfort of their own home [15,16], making peer support available to those who are unable to access such services in their local community. This is particularly important for people who are unable to travel or who feel uncomfortable attending in-person groups. Research shows that people with YOD use a variety of platforms for online peer support [17]. For example, discussion forums [18,19,20], Facebook [21], Twitter/X, [22], and videoconferencing platforms [23]. The wide variety of platforms means that people can choose a platform and mode of communication that meets their individual needs and preferences.

Patient and public involvement (PPI) consultations with people with YOD and consultations with online peer support facilitators indicated that clear guidance is needed to ensure that online peer support is safe, helpful, and engaging. Such a guidance should contain advice for facilitators on how to moderate and manage groups, as well as information about what online peer support groups can bring people with YOD. This work describes the development of a Best Practice Guidance on online peer support for people with YOD. It is part of a wider study that explored the following: (1) how people with YOD experience online peer support, (2) the benefits and challenges, (3) the barriers and how these could be overcome, and (4) how to optimize online peer support for people with YOD [24]. To develop the Best Practice Guidance we brought together evidence from systematic literature reviews [25,26,27], focus groups [23], an online survey [17], and interviews [28]. The aims of the Best Practice Guidance are the following: To provide people with YOD with evidence-based, relevant, and accessible information about what online peer support is and how it could help them;To provide providers of online peer support with guidelines on how to optimize the positive outcomes for people with YOD; andTo provide healthcare professionals with a concise and accessible tool for signposting.

## 2. Materials and Methods

This study received ethical approval from the London Bromley Research Ethics Committee (reference number: 21/LO/0248). The Best Practice Guidance consists of two parts: one for people with YOD on getting involved in online peer support (Part 1) and one for online peer support facilitators (Part 2). The aim of Part 1 is to give an overview of what online peer support entails, to address common questions and concerns, and to provide a list of resources where people can find more information. The aim of Part 2 is to provide clear guidelines for facilitators on how to optimize online peer support for people with YOD. Facilitators of online peer support can be health or social care professionals or people with lived experience of dementia.

### 2.1. Developing Draft 1 of the Best Practice Guidance

An overview of how the Best Practice Guidance was developed is presented in Figure 1. The Best Practice Guidance was developed through four different sub-studies and a consultation with members of the European Working Group of People With Dementia (EWGPWD). The methodologies and main findings of the sub-studies and the consultation with the EWGPWD are presented in Section 2. An overview of how each sub-study contributed to the Best Practice Guidance can be found in Section 3. To decide on a style and format of the Best Practice Guidance, we used examples from the Dementia Engagement and Empowerment Project (DEEP). These were chosen because they were developed by and for people with dementia. Senior members of the research team have extensive experience in working with and developing study materials for people with dementia. This further informed the format and style, for example, the font, font size, line spacing, and the use of color, bold text, and images.

#### 2.1.1. Systematic Literature Research

The systematic literature review aimed to obtain insights into best practices and challenges of online peer support. As the scholarly literature on online peer support for people with dementia was relatively limited, it was decided to conduct a systematic review on online peer support for people with Parkinson’s disease, multiple sclerosis (MS), and amyotrophic lateral sclerosis (ALS). These conditions were selected because they are also chronic and neurodegenerative in nature and prevalent among people under the age of 65, and people with these conditions can experience challenges and impacts similar to people with YOD [29,30,31]. The narrative synthesis methodology was used, following the steps by Popay, Roberts [32] for undertaking and reporting on narrative syntheses.

The systematic reviews generated insights into successful elements of online peer support (aspects that make online peer support work well for the people engaging with it), and potential risks and how to mitigate these [25,26,27]. The review on Parkinson’s disease emphasized the importance of similarity among peers and that people may find it easier to discuss sensitive topics (e.g., around relationships and intimacy) online rather than in-person [25]. The review on MS highlighted that simply reading about the experiences of others without actively interacting with posts can already generate feelings of social support. Furthermore, it was found that the feature of archiving posts can be helpful when trying to find posts related to a specific topic [26]. Finally, the review on ALS identified that online peer support can be particularly helpful for people with physical limitations for whom it may be difficult to leave the house independently. In particular, text-based or voice-to-text options are important for people who experience difficulties with speech and language [27]. A more detailed overview of the systematic review findings can be found in the associated publications [25,26,27].

#### 2.1.2. Focus Groups

The scholarly literature has mainly focused on online peer support for people with dementia in text-based platforms. However, through patient and public involvement (PPI) consultations with people with YOD and peer support facilitators, we found that, due to the COVID-19 pandemic, many in-person groups moved online using videoconferencing platforms. To explore how people with YOD experience peer support through videoconferencing platforms, four focus groups were conducted with existing peer support groups for people with YOD, which included 20 people with YOD. The focus groups took part on each group’s usual meeting platform. The group facilitators opened the meeting and welcomed the group members but were not present during the focus group. All groups were meeting online through videoconferencing platforms at the time of the focus group. A thematic analysis following the guidelines of Braun and Clarke [33] was conducted.

The focus groups generated insights into how people with YOD experience meeting with their peer support group through video meetings, what makes online peer support work well, the potential challenges and barriers, and how people cope with these. The findings show that people with YOD experienced social support and friendship through their online group, and they enjoyed the possibilities it offered them to meet others from across the country and internationally. People appreciated the audio–visual element, as they could hear and see the others, creating a sense of togetherness. However, some of the groups missed being together in person, and they noticed that not everyone from their in-person group managed to take part in the online meetings. This highlighted the problem of digital exclusion. A more detailed overview of the focus group findings can be found in the associated publication [23].

#### 2.1.3. Online Survey

Another gap in the scholarly literature was that there were no studies on the views and experiences of people with YOD who do not engage in online peer support or who stopped engaging. Therefore, we conducted an online survey collecting the views and experiences of people with YOD regarding online peer support. In total, 69 people with YOD took part. The survey had tailored questions for three groups: (1) people who currently use online peer support, (2) those who had used online peer support in the past but stopped using it, and (3) those who had never used online peer support. All participants were people living with YOD. The multiple-choice responses were analyzed in SPSS and Fisher’s exact test was used. The free-text responses were analyzed using the thematic analysis procedures of Braun and Clarke [33] The online survey provided insights into which platforms people use for online peer support, what they liked about it, what they disliked about it, and what challenges they faced. The survey showed that the main reason for not engaging with online peer support was that people did not know about it or where to find more information. Furthermore, the survey also showed that negative experiences or online peer support not meeting someone’s needs were reasons for people to stop engaging. A more detailed overview of the survey findings can be found in the associated publication [17].

#### 2.1.4. Individual Interviews

To obtain an in-depth understanding of why people are hesitant to engage in online peer support, how to better support people to get involved, and what makes online peer support work well, individual interviews with survey participants were conducted. In total, nine people who took part in the interview. In total, 56 survey participants expressed interest in being involved in an interview. Due to limited time and resources, a sample of 19 people was invited. This sample was selected considering the diversity and representativeness of the total sample, while also prioritizing people from less represented groups (e.g., ethnicity other than White British, people who live alone or who are in paid employment). Again, the procedures of Braun and Clarke [33] were used to conduct a thematic analysis of the interview data. 

The interviews provided more insights into topics for which the survey did not provide a deeper understanding. For example, participants shared some of the barriers that stopped them from engaging with online peer support, such as not knowing what to expect and feeling anxious to potentially see others who are in a more advanced stage of dementia. Participants suggested that having a detailed description of the group would be helpful. Participants also highlighted the importance of having a skilled facilitator or moderator. A more detailed overview of the interview findings can be found in the associated publication [28].

#### 2.1.5. Consultation with the European Working Group of People with Dementia (EWGPWD)

The Best Practice Guidance was discussed in a meeting with members of the EWGPWD to gather input on the content. In total, six people with dementia and six family members provided feedback. People received a summary of the project and guiding questions two weeks before the meeting. The questions addressed topics such as group size, how to support people with dementia in online meetings, and what kind of information would be helpful to include in the Best Practice Guidance. People also shared their own views and experiences beyond the questions.

### 2.2. Developing Draft 2 and the Final Version

The first draft of the Best Practice Guidance was sent to 60 people with YOD who took part in one of the studies or were involved as PPI members and 14 professionals working with people with YOD. The professionals included peer support facilitators, dementia advisors, academics, and healthcare professionals who were also involved in the initial consultations that informed this work and were identified through the professional networks of the research team. People were contacted via email, phone call, or post. The professionals included the facilitators of the peer support groups that took part in the focus groups and others who were identified through the network of the research team. Everyone received both versions of the Best Practice Guidance alongside an information letter, which included how the Guidance was developed, guiding questions to help people give feedback, and the contact information of the research team. The guiding questions can be found in Appendix A. People were reminded that they could share any feedback they wanted and did not have to stick to the guiding questions, and that they could choose if they wanted to give feedback for one or both parts. After receiving the PPI feedback, both parts of the Best Practice Guidance were adjusted accordingly. These adjustments were discussed within the research team, after which the final version of the Best Practice Guidance was developed.

## 3. Results

The final versions of both parts of the Best Practice Guidance can be found in Appendix A and are summarized in Table 1 and Table 2. An overview of the different sections, how they were developed, and what changes were made after the PPI feedback is presented in Table 3. Section 3 first describes how the first draft of Part 1 of the Best Practice Guidance was developed, followed by the development of the first draft of Part 2. This is followed by three sections that describe the feedback on the first draft of the Best Practice Guidance of people with YOD, professionals, and the research team.

### 3.1. Part 1: Guidance for People with YOD on Engaging with Online Peer Support

This guide is to inform people about different types of online peer support, what they can expect from it, how it can help them, and where they can find more information. It also addresses some common questions and concerns. The guidance consists of five sections: What different types of online peer support are there?What can I expect from online peer support?How can online peer support help me?How can I overcome technological challenges?Where can I find more information?

#### 3.1.1. What Different Types of Online Peer Support Are There?

This section was included because many people with YOD are unaware that online peer support exists [17]. Therefore, this section includes an overview of different platforms that can be used for online peer support and differentiates between text-based and audio–visual platforms. Some examples of text-based platforms include social media platforms, WhatsApp, and discussion forums. The pros of such platforms are that people can interact at their own pace and in their own time, and they can search for specific topics that are important to them [25,26,27]. However, a con can be that text-based platforms tend to be larger in nature and can feel more anonymous. This could make it more difficult for some people to feel a sense of connection [25].

Examples of audio–visual platforms include Zoom, MS Teams, and FaceTime. Research shows that through video meetings, people can build friendships and feel connected [17,23,28]. However, these meetings are on a specific day and time, and people may need in-the-moment support outside of these meetings [28]. Finally, this section in the Best Practice Guidance contains a message stating ‘Safety first!’, because the systematic reviews on online peer support for people with Parkinson’s disease and multiple sclerosis showed that safeguarding messages are common in text-based platforms [25,26].

#### 3.1.2. What Can I Expect from Online Peer Support?

Not everyone may be aware of what online peer support entails and that there are groups specifically for younger people [23]. Furthermore, some people feel hesitant to join (online) peer support groups. Reasons can be not knowing what to expect or assuming that it will not be helpful or suitable because of their age or because they experience less severe symptoms and, therefore, do not need as much support. Knowing what to expect and who the group is for (e.g., people of a certain age, people still working, people with a specific type of dementia) could help people feel more comfortable in trying online peer support [28]. Research shows it is important that people share similarities with other people in the group [25], and some people may find themselves trying out several groups before they find one that suits them [28].

#### 3.1.3. How Can Online Peer Support Help Me? 

Research shows that some people with YOD may not always be aware of what online peer support exactly entails and what it could bring them. Some people may feel they have enough support from family and friends and that, therefore, they do not need peer support [17,28]. However, peer support is unique and could offer additional benefits. Peers may be able to provide different support or insights and share mutual understanding and empathy because they have a shared experience of living with YOD. Additionally, through peer support, people can also learn about other support services [17,23,28]. These benefits can be associated with peer support in general. However, there are also unique benefits to peer support in online settings. For example, online platforms can overcome geographical barriers [17], and it can be helpful for people who are unable to travel or who do not feel comfortable attending in-person groups [23,28].

#### 3.1.4. How Can I Overcome Technological Challenges? 

For some people, technological challenges or difficulties they experience due to their dementia symptoms are barriers to online peer support [17,23,28]. This section aims to address some of the challenges that people with YOD in the focus groups and the interviews shared, and how they coped with these. This section is presented in a question-and-answer format, for example, ‘My dementia makes it difficult to use technology. What can I do?’ This is followed by a list of hints and tips from people with YOD who took part in one of the sub-studies.

#### 3.1.5. Where Can I Find More Information?

Our research shows that many people are unsure where to go for more information, either regarding (online) peer support, or support in general [17]. More specifically, finding the right information and support at the right time can be a long and difficult journey [23,28]. This section is divided into three parts addressing different information needs:I want to find a peer support group.I want to learn from other people’s experiences but not be part of a group.I want more information about young-onset dementia.

The resource lists are informed by an online survey and individual interviews with people with YOD and include organizations that people with YOD mentioned and found helpful [17,28]. The ‘Opening Doors’ organization did not come forward in either of the sub-studies but was identified through the professional network of the research team.

### 3.2. Guidelines for Facilitators

This guide is for those who facilitate online peer support groups to provide them with hints and tips, coming directly from people with YOD who are engaged in online peer support or who are interested in becoming involved about how to optimize online peer support and how to make it a positive experience for those involved. It was also informed by experiences and insights from peer support facilitators. This guide contains two main sections: one for peer support in video meetings, and one for peer support in text-based platforms.

#### 3.2.1. Peer Support in Video Meetings

This section is divided into four parts: (1) what is important before the meeting, (2) what is important during the meeting, (3) what is important after the meeting, and (4) further practical things to keep in mind. Each part contains hints and tips coming directly from people with YOD.

#### 3.2.2. Peer Support in Text-Based Platforms

This section is specific for moderators of online peer support groups on text-based platforms. Because some people reading the Best Practice Guidance may not be familiar with online peer support in text-based platforms and what their role can be, this section starts with general information about what online peer support in text-based platforms is. It includes the suggestion to consider offering a Q&A session with a healthcare professional. A systematic review on online peer support for people with Parkinson’s disease showed that people appreciated the opportunity to ask their questions directly to a professional, for example, regarding medication use [25]. Research involving people with YOD shows that people may have questions on how to live well with dementia after a diagnosis [17,28]. While peers can give practical hints and tips, some people may like having an opportunity to ask specific questions to a professional as well. However, not everyone may want this to be part of their peer support group, so it is always something that the facilitator or moderator should discuss with the group.

### 3.3. Feedback from People with YOD

Out of the 60 people with YOD that were contacted, 7 provided feedback, along with 1 additional person with dementia whose age was unknown (contacted by one of the professionals from their own network). An overview of the feedback and how it was addressed is presented in Table 3. Overall, people were positive about the Best Practice Guidance. Some shared tips on restructuring the content (e.g., putting a certain item at the top of a section). One person commented on the use of graphics hat seemed slightly childish. In the final version, the graphics have been adjusted. Most people shared that the guidance is clear and understandable, that the information included is relevant, and that the length is just right. One person said:

“It is full of good info. It probably took me 12 months to find a lot of the info but with this guide it would have been a lot quicker and easier”.

### 3.4. Feedback from Professionals

Out of the 14 professionals that were contacted, 5 gave feedback. An overview of the feedback and how this was addressed is presented in Table 3. They included two people from different dementia organizations, of whom one was also an academic, and three facilitators of (online) peer support groups for people with dementia and informal carers, of whom one was also a former informal carer of someone with YOD. Overall, the professionals were positive about both parts of the Best Practice Guidance. While the majority felt that the guides were clear and that the information included was relevant, two mentioned that the guidelines for facilitators may be too dense and confusing for people with dementia to read. Throughout the guidance, sentences were shortened where needed and made active instead of passive. One person was involved with an online discussion forum for people with dementia and family carers. They shared that volunteer hosts also welcome new members and help them get started on the platform, while moderators are more involved in the content (e.g., removing harmful messages) and making sure everyone follows the ground rules.

### 3.5. Feedback from the Research Team

The main point of feedback was instead of giving recommendations, reminding the readers that these are things that people with YOD shared and identified as important. Further comments were related to the use of language. For example, adapting user-friendly language by saying ‘we,’ avoiding vague terms such as ‘it can be a bit more anonymous’ and generally being more explicit (e.g., replace ‘you can read about …’ with ‘this section summarizes …’). Finally, it was recommended to include an acknowledgement section that explains how the Best Practice Guidance was developed and includes the contact details of the research team.

## 4. Discussion

The Best Practice Guidance offers people with YOD with evidence-based, relevant, and accessible information about what online peer support is, how it could help them, and hints and tips on how to navigate (technological) challenges. Throughout the different sub-studies, the research team worked closely with people with lived experience of YOD through PPI and participation in studies. This helped in developing a first draft of the Best Practice Guidance that reflected people’s views and experiences and that needed only minor changes. During the COVID-19 pandemic, many health and social care services for people with dementia moved online. This was often a difficult process, where everyone involved had to learn while doing, without specific guidance in place on how to navigate the transition from in-person to online [34]. Consultations with peer support facilitators for this study confirmed that this was the case for peer support groups for people with YOD as well. Groups noticed that some members were unable to join the online meetings and, as a result, missed out on the support and benefits [23]. Taking part in online peer support can be challenging for people with dementia. Due to the nature of symptoms, this may be even more so for people with rare forms of dementia [35], which are relatively more common among younger people [36,37]. The Best Practice Guidance includes unique hints and tips, directly from people with YOD, on how to cope with different symptoms that may make it more difficult to use technology or engage in online communication. 

Furthermore, the Best Practice Guidance offers information and hints and tips for group facilitators or moderators on how to support people with YOD in accessing and engaging with online peer support. The Best Practice Guidance can support organizations offering online peer support, or those that want to get started with this, with clear and accessible guidelines. It also offers a place for organizations to advertise their online peer support groups. Finally, it can be a way for healthcare professionals to signpost to online peer support. Research shows that people with YOD often experience either a lack of information at the time of diagnosis or information overload. When people do receive information, it is not always relevant at that time [38]. Online peer support can be a lifeline for people with YOD and give them hope and purpose again [23]. Nevertheless, research showed that only a minority found out about peer support through their doctor, while for most, it was a long journey to find it themselves [17]. The Best Practice Guidance addresses this gap by providing healthcare professionals with a concise and accessible document that they can share with people with YOD.

### 4.1. Limitations

One of the main challenges was to involve more people from underrepresented groups (e.g., people from ethnic minorities, people living alone, or people who are still in paid employment). Therefore, the views and experiences of these groups may not be represented sufficiently in the Best Practice Guidance. Furthermore, there was only a relatively small group for the PPI feedback on the draft of the Best Practice Guidance despite the invitation to provide feedback being sent out to sixty people with YOD and fourteen professionals. 

The Best Practice Guidance was not tested in a real-world setting. For example, it was not disseminated in healthcare settings where people with YOD who were not involved in the study could receive them and give feedback on whether the Guidance was relevant and helpful for them. Finally, the guidelines were also not further disseminated among facilitators and moderators, for example, through dementia organizations, to see if they found the guidelines helpful and whether anything was missing.

### 4.2. Recommendations for Future Research

Future work may focus on implementing the Best Practice Guidance in practice to make it freely accessible for people with YOD, their families, and professionals. An online survey showed that the most popular resources for information on (peer) support services in the UK are dementia organizations [17], suggesting that these are key organizations to reach people with YOD. Despite the wide variety of dementia organizations and available resources, the current work also found that there is a lack of awareness of online peer support among people with YOD, and that support services, including online peer support, were difficult to find [17,23]. Grey literature [39] or content analysis [40] methods could be useful in generating an overview of online peer support services for people with YOD. Finally, future research should focus on how to implement and sustain online peer support for people with YOD. This research could follow the example of the implementation research for the Meeting Centre Support Program, which includes community-based meeting centers for people with dementia and their families [41].

## 5. Conclusions

The Best Practice Guidance on online peer support for people with YOD bridges the gap between research and practice. It collates research findings in a concise and accessible format and provides (1) people with YOD with evidence-based, relevant, and accessible information about what online peer support entails and how it could help them, (2) providers and facilitators with guidelines on how to optimize online peer support for people with YOD, and (3) healthcare professionals with a tool for signposting. Future research is needed to implement and disseminate the Best Practice Guidance among dementia organizations and healthcare practices to monitor whether it is adopted in practice and to identify any further gaps and information and support needs regarding online peer support for both people with YOD and health and social care professionals.

## Figures and Tables

**Figure 1 behavsci-14-00746-f001:**
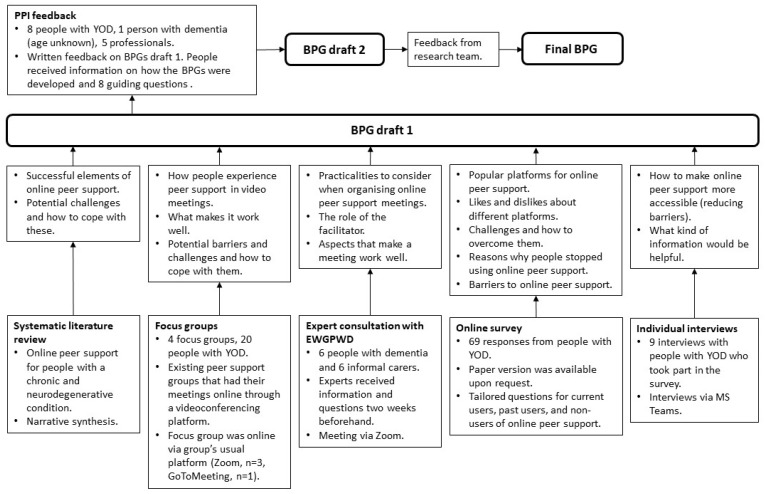
Development of the Best Practice Guidance (BPG).

**Table 1 behavsci-14-00746-t001:** Summary of Best Practice Guidance Part 1.

Best Practice Guidance Part 1: Engaging in Online Peer Support for People with YOD	
Section	Summary
What different types of online peer support are there?	Platforms using text and writing (e.g., social media platforms, WhatsApp, and discussion forums). Positives include that you can read posts at your own convenience and pace. Negatives include that you cannot see other people. Platforms using spoken language (e.g., Zoom, FaceTime). Positives include that you can see other people. Negatives include that you have to meet at a specific day and time.
2. What can I expect from online peer support?	Peer support is friendly and non-judgmental. You do not have to say anything if you do not want to. It is ok to just observe. Different groups can have different goals and identities. Find a group that matches your needs and preferences.
3. How can online peer support help me?	Meet new people who go through similar things to you. You can learn from their experiences and they from yours. Online peer support can be particularly helpful if there are no in-person groups in your local area or if you are unable to travel. You can turn off your camera or mute yourself at any time if you need to.
4. How can I overcome technological challenges?	Know how to reach the group’s facilitator. They can help you if you have trouble accessing the group. Tell the group facilitator if you have any specific symptoms that make online meetings more challenging for you or if you would like support.
5. Where can I find more information?	Links to further information if people want the following: To find a peer support group; To learn from other people’s experiences but not join a group; More information about dementia.

**Table 2 behavsci-14-00746-t002:** Summary of Best Practice Guidance Part 2.

Best Practice Guidance Part 1: Engaging in Online Peer Support for People with YOD	
Section	Summary
Peer support in video meetings	
What is important before the meeting?	Get to know the person well before they join the group. Find out what their expectations, needs, and wishes are. Establish ground rules with the group and repeat and revise these regularly. Send out timely reminders and be available to provide support.
2. What is important during the meeting?	Give everyone a chance to speak. Allow the group to discuss what is important to them (either through a commonly agreed upon agenda or on the spot).
3. What is important after the meeting?	Check in with people afterward if they left the meeting suddenly or appeared distressed. Follow up with any notes or answers to questions.
4. Further practical things to keep in mind	Keep the group size and duration of the meeting manageable. Try to offer meetings on different days of the week and different times of the day to accommodate to different needs.
Peer support in text-based platforms	Make a clear statement on the group’s purpose and whom it is for. Make the group closed so that you as a moderator need to give permission for people to join. Welcome new members. Monitor the content, remove inappropriate content, and if possible, contact the author of the inappropriate content. Create a dedicated space to save resources and discussion topics if the platform allows that.

**Table 3 behavsci-14-00746-t003:** Best Practice Guidance key points, sources, and changes.

**Best Practice Guidance Part 1 for People with YOD on Engaging with Online Peer Support**				
**Section**	**Source**	**Key Points Draft 1**	**PPI Feedback on Draft 1**	**Changes for Draft 2**
What different types of online peer support are there?	Online survey [17]. Systematic reviews [25,26] for text-based platforms; focus groups [23] for audio-visual platforms; online survey and interviews [28] for both. Systematic reviews.	List of different audio–visual and text-based options used by people with YOD. Pros and cons for each type. Online safety message.	From people with YOD: Bullying or conflict, impact on mental health.	PPI In the ‘Safety first!’ box: *Be mindful of how online peer support affects your mental health. If you notice a negative impact, reach out to the group’s facilitator or moderator. You can also take a break from it or try finding another group. You can also unfollow someone that has a negative impact*.
What can I expect from online peer support?	Focus groups; interviews	What can online peer support be like? Common questions and concerns about online peer support.	No feedback	No changes
How can online peer support help me?	Focus groups; online survey; interviews. See above (1). Systematic review on MS [26]; interviews.	Benefits of peer support. Benefits of online. ‘Did you know? Even just listening or reading about others’ experiences can be helpful.’	No feedback	No changes
How can I overcome technological challenges?	Focus groups, interviews.	Question-and-answer format of challenges people with YOD experienced and how they coped with them.	No feedback	No changes
Where can I find more information?	Focus groups; online survey; interviews. Online survey; interviews; systematic review on MS. Focus groups; interviews.	I want to find a peer support group. I want to learn from other people’s experiences but not be part of a group. I want more information about YOD.	No feedback	No changes
**Best Practice Guidance Part 2 for Facilitators**				
**Section**	**Source**	**Key Points Draft 1**	**PPI Feedback on Draft 1**	**Changes for Draft 2**
Peer support in video meetings	Informal consultation with peer support facilitators; consultation with EWGPWD; focus groups. Consultation with EWGPWD; consultation with peer support facilitators; interviews. Consultation with EWGPWD; interviews. Consultation with EWGPWD.	Important things before the meeting. Important things during the meeting. Important things after the meeting. Further practical things.	From people with YOD: How can I identify a group that matches my needs? Bullying or conflict. Staying in contact outside the meetings.	Under ‘Further practical things’: *Make sure to have a clear description of the group. This should include information on who the group is for, what generally happens during the meetings and what kind of topics are discussed, and when and on which platform the group meets.* A brief version of this point has been added to the key points in the same section. Under ‘What is important during the meeting’: *Make sure the meeting is a safe and confidential space for everyone. Speak up against inappropriate or harmful comments and bullying.* Under ‘What is important after the meeting’: *If there were any inappropriate, disrespectful, or harmful comments or if you noticed bullying during the meeting, address this. Contact both the person who made the comments and the person who received them.* Under Section 4 ‘Further practical things’: *Ask the group how they feel about sharing contact details with each other so that they can stay in contact outside of the meetings if they want to. Make clear that this is optional and that no one should feel pressure to do so.*
Peer support in text-based platforms	Consultations with peer support facilitators; online survey. Systematic reviews; online survey; interviews. Systematic reviews; interviews.	Different text-based platforms Group description and accessibility Role and expectations of the moderator	From professionals: Role of host/moderator, welcoming people	Under ‘Your role as a moderator/facilitator’: *Welcome new members and explain how the group works.*

## Data Availability

The data used in the study can be made available upon requests addressed to the corresponding author.

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
