# Peer review of "Development of a Best Practice Guidance on Online Peer Support for People with Young-Onset Dementia"

_behavsci, 2024, doi:10.3390/bs14090746_

Round 1

Reviewer 1 Report

Comments and Suggestions for Authors

Thank you for an opportunity to review your paper. Best practice guidelines for peer support are a welcomed contribution to the literature. They are relevant for both lay facilitators who have no knowledge of groups and for practitioner facilitators who have no knowledge of group facilitation with people living with dementia. Part 1 and Part 2 guidelines also appear to be equally important and I appreciate that you've done just that. I have a few comments for your consideration - much of which would help with clarity of the information presented:

1. In your introduction you state, "Feedback from people with YOD and facilitators of online groups has indicated that clear guidance is needed...." Can you clarify what feedback you are referring to?

2. Developing draft 1 of the guidance: The first paragraph here doesn't introduce the reader to Figure 1 and the related explanations of the different methods used to collect data. I find it unusual (also confusing) for the authors to describe 5 data collection methods but do not include how the samples were obtained (e.g., online survey), only very briefly present some of the findings, and provide no explanation of how the findings were then integrated to become guidance. Are the guidance sections and explanations in sections  3.1 and 3.2 based on the findings from all sources? I am also thinking here, for example, about how any disagreements or opposing views were dealt with when the guidance was being prepared. Might some of the findings or analyzed data be included as a supplementary file?  

3. I am not clear as to who constitutes an "expert". People living with dementia and family members are considered an expert in the instance of the EWGPWD. However, people living with dementia and family members were not otherwise "experts" for other data collection purposes?

4. The various samples are not described. It would be interesting to know, for example, the characteristics of people (type of dementia, for example) who do not use online peer support versus those who do.  Were the 9 who participated in the interviews the total number of volunteer participants from the survey?

5. For the focus groups you had 20 people take part from 4 groups.  Was that 5 people in each of the four groups?  Given the small group size may this reflect the type of feedback received? Were the group facilitators part of the focus group or just part of the draft 2 process?

6. Developing draft 2 and final version: Who were the 14 professionals contacted and why these 14 people?

7. 3.3 indicates another 8 people living with YOD gave feedback, but it not clear where they came from. 3.4 identifies 5 professionals giving feedback, and again, it is not clear who they were recruited.  You refer to "majority" a couple of times and does mean 5/5?

8. In the conclusions you appropriate identify issues around dissemination, uptake and effectiveness. Is this being planned by the team?

Reviewer 2 Report

Comments and Suggestions for Authors

You have to modify the abstract, these are the guidelines:

Better describe the Methodology:

 - for example "The guideline was developed through a systematic literature review, focus groups, online surveys, interviews and consultations with patients and the public."

Summarize Results:

 - Guidelines were identified for both health personnel and people affected by YOD.

Conclusions and Implications

 emphasize that more scientifically rigorous studies should be promoted for its implementation

Introduction

Contextualization give more details Explain in more detail the importance of virtual peer support for people affected with YOD.

Expand the Literature Review

 - Include more previous studies if they exist that have explored virtual peer support for YOD

Present more organized results

 - organize the results into sections such as participant perception, benefits, challenges and recommendations.

Reviewer 3 Report

Comments and Suggestions for Authors

Thank you for giving me the opportunity to read your manuscript, which clearly has a lot of work behind it and does a very nice job of bringing together information from many sources. Therefore, I missed is a bit more reflection on your way of bridging the gap between research and practice (combining information from various sources, the value and role of PPI, etc.). And another little thing that for some reason confused me: The aim of part 2 is to provide guidelines for facilitators and I understood that summary of these are presented in table 2. However, there are tips in point four of the table (further practical things to keep in mind) that seem to be more relevant to participants. For some reason I couldn't figure out why.

Round 2

Reviewer 2 Report

Comments and Suggestions for Authors

The authors have made the requested modifications

Author Response

Thank you for taking the time to review our manuscript for both rounds and for helpful suggestions.